# Caring for Mothers: A Narrative Review on Interpersonal Violence and Peripartum Mental Health

**DOI:** 10.3390/ijerph18105281

**Published:** 2021-05-16

**Authors:** Marianna Mazza, Emanuele Caroppo, Giuseppe Marano, Daniela Chieffo, Lorenzo Moccia, Delfina Janiri, Lucio Rinaldi, Luigi Janiri, Gabriele Sani

**Affiliations:** 1Department of Psychiatry, Fondazione Policlinico Universitario “Agostino Gemelli” IRCCS, 00168 Rome, Italy; giuseppemaranogm@gmail.com (G.M.); lorenzomoccia27@gmail.com (L.M.); delfina.janiri@gmail.com (D.J.); luciorinaldi.psi@gmail.com (L.R.); luigi.janiri@unicatt.it (L.J.); gabriele.sani@unicatt.it (G.S.); 2Università Cattolica del Sacro Cuore, 00168 Rome, Italy; danielapiarosaria.chieffo@unicatt.it; 3Department of Mental Health, Health Local Unit ASL ROMA 2, 00159 Rome, Italy; emanuelecaroppo@gmail.com; 4Service of Clinical Psychology, Fondazione Policlinico Universitario “Agostino Gemelli” IRCCS, 00168 Rome, Italy

**Keywords:** depression, perinatal women, postpartum, pregnancy, violence, personalized medicine

## Abstract

Interpersonal violence in the perinatal period is frequent and should be considered a prominent health issue due to the risk of escalation of violence and the significant impact on mothers’ parenting after childbirth. Domestic violence during pregnancy can be associated with fatal and non-fatal adverse health outcomes due to the direct trauma to a pregnant woman’s body and to the effect of stress on fetal growth and development. Emotional violence is a risk factor for prenatal and/or postpartum depression. Recent studies focusing on abusive situations during peripartum and possible preventive strategies were identified in PubMed/Medline, Scopus, Embase, and ScienceDirect. All of the available literature was retrospectively reviewed with a special attention to peer-reviewed publications from the last ten years. Results of the present narrative review suggest that perinatal health care professionals (general practitioners, gynecologists, obstetricians, psychologists, psychiatrists) should promptly detect interpersonal violence during and after pregnancy and provide health care for pregnant women. It seems pivotal to guarantee psychological care for abused women before, during, and after pregnancy in order to prevent the risk of depressive symptoms, other mental or physical sequelae, and mother-to-infant bonding failure. There is an urgent need for multifaceted interventions: programs should focus on several risk factors and should design tailored care pathways fitted to the specific needs of women and finalized to support them across the lifespan.

## 1. Introduction

A large body of literature supports that intimate partner violence—physical, sexual, psychological, or emotional abuse by a partner—is detrimental for women’s health. Psychological abuse is a type of abuse involving the use of verbal and social tactics in order to control someone’s way of thinking. Abusers will often convince the victim that they are crazy, manipulate them, or make harmful threats towards them. Emotional abuse is a type of abuse that involves controlling someone’s feelings and causing intense mental trauma [1]. Intimate personal violence occurs frequently all over the world, with prevalence ranging between and within countries from 15 to 71% [1]. Across the globe, the prevalence of emotional violence to pregnant women is estimated to range from 8% to 78% and physical violence from 4% to 39% [2]. Domestic violence is often directed at women, frequently occurring within the context of marriage or long-term relationships. Episodes of intimate partner violence in the household may create an adverse environment both for the caregiver experiencing the abuse and for children who are witnesses of the violence. In addition, experiences of violence usually compromise the ability and quality of parenting of the traumatized subject. Such events as conception, pregnancy, childbirth, and early transition to parenthood should usually be considered happy and joyful, but they could also represent new challenges to mothers, fathers, and, generally, to couple relationships [3]. In fact, in addition to the physical and hormonal changes due to pregnancy, it should not be overlooked that emotional, social, and economic adjustments may cause stresses and affect individual’s and couple’s coping strategies, possibly leading to the initiation, continuation, or increased frequency or severity of psychological and physical aggression [3]. The situation may be further complicated by additive stressors with consequent cumulative effects: this is the case of partners that are separated or suffer from mental illness or substance abuse, or the case of a coerced or unwanted pregnancy. Emotional and physical intimate partner violence in pregnant and postpartum women increases the risk of incident homelessness, with effects that are independent from and comparable to, or even greater than, the risk of financial adversity [4]. There is no doubt that pregnancy represents a developmental transition, and as a substantial change, it includes potential buffering effects derived from psychosocial resources (for example, women’s relationship power and partner social support), but it could also increase the levels of relationship stress with a possible association or escalation of intimate partner violence before and after a child’s birth [5]. There is evidence that intimate partner violence represents one of the most common health risks in the perinatal period (from pregnancy through 1 year postpartum encompassing the antenatal, intrapartum, and postpartum periods), with higher rates reported in women who are unmarried, young, separated, nonwhite, and poorly educated [3]. It has been evidenced that intimate partner violence during pregnancy contributes to maternal mortality from pregnancy-associated deaths (i.e., a maternal death that is attributable to a condition that is unaffected by the pregnancy and occurs within 1 year of the pregnancy, such as homicide, suicide, and drug-related overdose) [6]. Pregnancy-associated feminicide is responsible for a mortality burden comparable to the leading specific obstetric causes of maternal mortality, and intimate partners are the most common perpetrators of these homicides. In the USA, black women experience pregnancy-associated intimate partner homicide victimization at a rate 8.1 times greater than their non-pregnant peers [7].

It is often difficult to identify situations of intimate partner violence since it is mostly consumed in private and victims are afraid of disclosing abuse. What is worse is the fact that violence frequently happens in contexts where legal systems and cultural norms do not treat these actions as a crime [8]. It is reported that there is a frequent direct relationship between being a victim of violence and socioeconomic vulnerability and cultural factors: socioeconomic, sociodemographic, and behavioral characteristics frequently contribute to the risk of being both a perpetrator and a victim of violence against women during pregnancy. It seems that psychological violence by an intimate partner is the most prevalent among pregnant women, and that women experience violence more frequently during pregnancy if they are younger, have a lower income and less schooling, if they start their sexual life before the age of 14, and if they wish to interrupt pregnancy [9].

Another important issue is that for one quarter of women who experience domestic and family violence, the abuse begins during pregnancy. When violence was previously present, it tends to escalate in frequency and severity during pregnancy and early motherhood. There is a higher risk in younger mothers (between the ages of 18 and 24 years). Many adverse outcomes are associated with domestic violence during pregnancy: higher rates of preterm and low birth weight, poor reproductive health, unintended pregnancy, planned and spontaneous abortion, postnatal depression, and substance abuse [10]. A large population-based cohort study on more than 2 million mother–baby pairs in the UK showed that pre-pregnancy psychosocial risk factors (i.e., previous birth before 20 years of age, hospital contacts related to adversity or mental health or behavioral conditions, and deprivation) were associated with substantially increased risks of low birth weight, preterm birth, injury, and death during the 12 months from postnatal discharge [11].

A recent meta-analysis showed that the prevalence of any antenatal depression is 20.7%. Additionally, 15% of pregnant women experienced major antenatal depression. There are eight common factors associated with antenatal depression: history of depression, lack of social support, experience of violence, single/separated/divorced status, unemployment, unplanned pregnancy, history of smoking, smoking during pregnancy [12].

These results indicate that the mental health of pregnant women should be paid more attention and demonstrate a need for effective interventions before, during, and after pregnancy to reduce the downstream burden on health services and prevent long-term adverse effects for children.

## 2. Methods

Studies focusing on abusive situations during peripartum and possible preventive strategies were identified in PubMed/Medline, Scopus, Embase, and ScienceDirect. We searched cited databases for peer-reviewed publications related to the following keywords: “peripartum”, “intimate partner violence”, “interpersonal violence”, “domestic violence”, “pregnancy”, “postpartum depression”, “postpartum anxiety”, “perinatal women”, “perinatal health care”, on 1 February 2021. Inclusion criteria included original studies in peer-reviewed journals focusing on intimate partner violence during peripartum. Eligible studies had to report data on victims of intimate partner violence regarding prevalence, clinical course, or psychopathology. Both longitudinal and cross-sectional studies were admitted and could be retrospective or prospective. There were no time limits or language limits in regard to the selection of appropriate studies. Reviews or meta-analyses focusing on interpersonal violence during peripartum and possible preventive strategies were also consulted. The latter were used to search among their references for possible further eligible studies. Studies were excluded if they did not focus on or were unrelated to the subject matter, or if they were case reports or series. All authors participated in the selection of eligible studies to include in the present review. Our search strategy on the aforementioned date produced a grand total of 447 articles to assess. We retrospectively reviewed all of the available literature and decided to focus on more recent and complete publications with a special attention to works from the last ten years (from 2011 to 2021). In particular, we considered a total of 58 publications focusing on the link between intimate partner violence and prenatal and/or postpartum depression, the risk of suicide for abused mothers, and preventive measures to protect mothers from violence before, during, and after pregnancy.

## 3. Results

The present narrative review is a comprehensive, critical, and objective analysis of the current knowledge on the important topic of interpersonal violence and perinatal mental health, with a special attention to the more recent literature. The considered studies agree in affirming that women belonging to marginalized communities, women with lower incomes, or women with less education are at greater risk for experiencing intimate partner violence during pregnancy [13,14,15,16,17,18,19]. Moreover, intimate partner violence represents a potential trigger for the development or worsening of prenatal and/or postpartum depression, and, on the other hand, peripartum depression may heighten the risk for intimate partner violence [20,21,22,23]. There is agreement that the mental health of pregnant women should be paid more attention [12]. The risk of suicide is a major concern for depressed mothers in the perinatal period [24,25,26,27,28]. As for preventive measures to protect mothers, all considered studies agree in emphasizing the importance of providing access to perinatal care to early detect and address domestic violence [5,26,29]. The most successful strategies to provide health care for pregnant women and to guarantee psychological care for abused women before, during, and after pregnancy thus far are represented by interventions of home visitation, sessions of supportive and family-based counseling, or interpersonal psychotherapy [20,30]. There is debate on the hypothesis that pregnancy could be a trigger for intimate partner violence, and this point needs to be clarified by future research [16], and another unmet need regarding perinatal mental health is in-depth study and attention to countries with diverse income levels [12]. The results of the present review show that interpersonal violence against pregnant women represents a pivotal perinatal health issue due to the high risk of escalated abuse and its great impact on mothers’ parenting after childbirth. All considered studies agree in recommending that perinatal health care providers (in particular, general practitioners, gynecologists, obstetricians, psychologists, psychiatrists) identify and give necessary support and referral for women experiencing domestic violence during peripartum, and for their children. The focus on maternity and early childhood services is strongly required and needs to be integrated and coordinated with adequate efforts to identify women at risk or victims of both present or past interpersonal violence and to support them across the lifespan. Preventive measures to protect victims of interpersonal violence during peripartum have been identified and should be considered a highest-priority issue for perinatal health care professionals (Table 1).

### 3.1. Intimate Partner Violence and Prenatal and/or Postpartum Depression

Women in their childbearing years have the highest overall rates of intimate partner violence, including during pregnancy. Studies indicate that women who experience intimate partner violence before pregnancy have a greater risk for becoming victims of abuse during and after pregnancy, as well. It has also been stressed that women belonging to marginalized communities, women with lower incomes or less education, and women of color may be at greater risk for experiencing intimate partner violence [13]. There is a peculiar, concerning form of physical violence during pregnancy: when the abusive partner hits a woman’s abdomen, there is not only the consequence of hurting the woman but also a great risk of potentially jeopardizing the pregnancy [14]. Harm to physical health experienced by intimate partner violence survivors includes acute and often visible injuries such as bruises, lacerations, fractures, and sight and hearing damage, in addition to chronic conditions such as hypertension, irritable bowel syndrome, fibromyalgia, asthma exacerbation, and chronic pain syndrome [15]. Moreover, victims of intimate partner violence during pregnancy are at great risk of being killed by the violent partner. There is debate on the hypothesis that pregnancy could be a trigger for intimate partner violence. In the World Health Organization multi-country study on women’s health and violence against women, the majority of women who reported physical abuse during pregnancy have also been beaten prior to getting pregnant, although there are many cases of domestic violence occurrring for the first time during a pregnancy. In addition, intimate partner violence has been confirmed as a consistent and strong risk factor for unintended pregnancies and abortions across a variety of settings [16].

Intimate partner violence represents a potential trigger for the development or worsening of prenatal and/or postpartum depression, and, on the other hand, peripartum depression may heighten the risk for intimate partner violence. Globally, severe postnatal-onset depression rates are three times higher than in other periods of women’s lives.

The current COVID-19 pandemic and restrictive measures with consequent serious social isolation and disruption of daily habits have amplified both the risk of domestic violence and psychological distress, and the risk of mood disorders in childbearing women [17,18]. In fact, it has been outlined that the prevalence rates of mental disorders among pregnant and postpartum women during the COVID-19 pandemic have been high, especially among multigravida women and women in the first and third trimesters of their pregnancy [19]. Living in daily exposure to violence and abuse may cause sadness and distress and may affect women’s perceived mental health profoundly. The mother’s relationship with their intimate partner has the greatest impact on their accomodation during pregnancy. In general, women who experience intimate partner violence are more likely to delay entry into prenatal care and develop symptoms of depression, anxiety, post-traumatic stress disorder, suicide attempts, and other mental health conditions, often complicated by poor maternal nutrition and use of tobacco, alcohol, and illicit drugs [20]. As for adverse health behaviors, it has been hypothesized that women smoke, drink, and take drugs notwithstanding their condition of pregnancy for self-medication to cope with the stress, shame, and suffering caused by the abuse [21]. It is difficult to ascertain an estimated prevalence of substance or alcohol use and smoking due to stigma associated with self-reporting substance abuse, drinking, and smoking and intimate partner violence. Some studies confirm that interpersonal violence significantly increases these behaviors in pregnant women and suggest a possible combination of chronic stress and substance misuse as a stress-coping mechanism that inevitably predisposes vulnerable women to unhealthy and dangerous behaviors [15].

Experiencing intimate partner violence during pregnancy increases by three-fold the odds of developing postpartum depression [22]. A systematic review was conducted to synthesize the empirical literature on the associations between intimate partner violence exposure (before pregnancy, during pregnancy, postpartum) and post-traumatic stress and depression symptoms in the perinatal period. In general, physical, sexual, and psychological violence forms were independently associated with perinatal depression and post-traumatic stress disorder [23]. Perinatal post-traumatic stress disorder and depression confer an increased risk of adverse pregnancy and child outcomes, even at subclinical levels.

Mental and physical health consequences related to interpersonal violence may persist long after the violence has ended. The coexistence of intimate partner violence and peripartum depression often results in negative effects on the health and wellbeing of both women and their children: intimate parner violence during pregnancy can lead to preterm labor, low-birth weight infants, fetal trauma, and, for some, neonatal and infant mortality [21]. In women hospitalized for physical assault, sexual assault, and intimate partner violence, it has been observed that violence was associated with adverse birth outcomes regardless of the timing or frequency of violence hospitalizations. This evidence may be related to previous research demonstrating that victims of violence frequently carry a history of abuse or other stressful adverse experiences that accumulate over the lifetime [24].

On the other hand, giving bith to a preterm or low-birth weight infant can increase a mother’s risk of developing postpartum depression and can have negative consequences on the mother–child bond, with potential longer-term effects on parenting and child development [25]. Bonding failure is defined as the experience of a mother of negative maternal feelings towards their infant, such as reduced affection, rejection, indifference, hostility, anger, and even impulses to harm. Additionally, depressed mothers are often preoccupied with their own relationship stress and mental health and report less maternal involvement, and ineffective parenting, which may contribute to a difficult infant temperament [1]. A vicious circle is created in which intimate partner violence during pregnancy is linked to mother-to-infant bonding failure, which in turn may led to abusive parenting behaviors by mothers. A possible explanation is the fact that intimate partner violence perpetrators often forbid pregnant women to express their affection and care for their baby and make it difficult to take close care of their newborn after childbirth, due to dynamics of jealousy and competition with the infant. Another possible reason for bonding failure could be that pregnant women who experience violence may feel difficulties in adapting to the new role as a mother [26]. Particularly in low-income and populous countries, intimate partner violence may be associated with an increase in neonatal mortality due to reduced postnatal health practices. Women who experience domestic violence are less likely to engage in maternal and child health protective behaviors, including health care seeking, prior and after pregnagncy. Interpersonal violence can compromise women’s health practices indirectly, by inducing stress/anxiety and depressive symptoms, or directly, since abusive male partners may hamper a mother’s protective health behaviors [27]. It has been shown that intimate partner violence is associated with a lower likelihood of exclusively breastfeeding infants in the first six months of life. Intimate partner violence may affect breastfeeding directly, through sore nipples and difficulty in relaxing for adequate let-down, but also indirectly, through lack of support or depression, self-doubt, body negativity, and anxiety [28]. In a sample of 779 mothers, psychological intimate partner violence was reported by one in five women, and the results show that this type of violence during peripartum, on average, doubles the avoidance of breastfeeding [29].

A history of physical and sexual intimate partner violence is associated with not using contraception postpartum [30]. Several maternal mental health problems are combined with perinatal intimate partner violence: depression, anxiety, psychosis, post-traumatic stress disorder, inability to trust others, self-harm, risky behaviors, and multiple psychosomatic conditions including chronic pain, and cardiovascular and metabolic conditions [3,31]. The primacy of postpartum depression has been widely demonstrated as a critical maternal characteristic impinging on parenting sensitivity. In addition, mothers with comorbid depression and post-traumatic stress disorder show the highest bonding problems with their infants [32]. A recent study on a large sample of women demonstrated that both recent and past exposures to intimate partner violence are associated with poor maternal physical and mental health 10 years after the birth of a first child. The evidence that women with past exposure to violence remain at higher risk for a wide range of mental and physical health problems illustrates the high personal costs and longer-term consequences of interpersonal violence for women’s health [33]. Moreover, it has been shown that mothers who were victims of intimate partner violence during pregnancy were more likely to have depression symptoms when their children turned 3, and maternal depression symptoms could directly predict children’s depression symptoms at age 15. Meanwhile, maternal depression symptoms could indirectly increase adolescent depression symptoms via physical punishment at age 5 and bullying victimization at age 9. Interventions designed to support pregnant women experiening intimate partner violence should explicitly address the intergenerational effects of violence and prevent and reduce the chances of adolescents to develop depression following mothers’ exposure to violence [34]. Another significant aspect regards the effects of childhood threat (physical, sexual, and emotional abuse and exposure to violence) and deprivation (physical and emotional neglect and separation from primary caregivers) experiences on pregnant women’s emotion disregulation, with an elevated risk of post-traumatic stress, negative emotional symptoms, and low perceived social support for future mothers [35]. A child’s exposure to violence in the family is a major predictor of intimate partner violence victimization later in life [36].

### 3.2. Risk of Suicide for Abused Mothers

The risk of suicide is a major concern for depressed mothers in the perinatal period, and in several high-income countries, suicide has been identified as one of the most common causes of death among women within 1 year after the end of pregnancy. Above all, women who are victims of intimate partner violence are at elevated risk for suicidal ideation and suicide attempts [37]. Suicidality is distressing to the mothers experiencing it, and it could undermine parenting and child development. There is a prevalence among mothers living in poverty of social risk factors for suicide and mental health problems, including violence, social isolation, stress, and low resources. It seems that mothers who completed suicide perinatally used more lethal methods than women at other stages of life [38]. A history of suicide attempts is associated with childhood trauma history and psychosocial impairments in low-income depressed mothers in home visits and may be related to the emotional and behavioral dysregulation caused by experiences of trauma and violence [39]. It has been demonstrated that social support can be an important protective factor and is inversely related to suicidal ideation over the first 18 months postpartum [40].

Suicide deaths are a leading cause of maternal mortality in the US. In a cross-sectional study of US childbearing individuals, the prevalence of suicidal ideation and intentional self-harm occurring in the year preceding or following birth increased substantially over a 12-year period. Racial ethnic minority individuals, those with a low income, and younger individuals, as well as those with comorbid anxiety, depression, or other serious mental illnesses, had larger escalations. Mood disorders represent a key risk factor for mothers’ suicide in the year prior to and following their children’s birth. Medication discontinuation, lack of ongoing treatment, and intimate partner violence are further risk factors for suicide among subjects with mood disorders. It seems imperative to provide clinical and policy interventions for addressing this severe maternal mortality crisis, tailored to meet the unique needs of childbearing women in the year before and following partum, particularly among high-risk groups [41]. The fact that, in the presence of violence, women and men become depressed is not surprising, but the most vulnerable group is women, especially in low-income countries; therefore, there is a need to train health workers and to offer countinuous effective assistance to pregnant women [42]. An epidemiological study focusing on maternal suicide in Italy showed that the majority of women who died by maternal suicide between 2006 and 2012 had a psychiatric history or a previously diagnosed mental disorder (with bipolar disorder and major depressive disorder reported as the more represented diagnoses) not registered along with the index pregnancy obstetric records and evidenced a frequent use of violent methods and a higher risk of suicide in the first weeks after giving birth. These data are consistent with previous findings in other maternal mortality surveillance systems worldwide [43,44] and demonstrate that the continuity of care between primary care, mental health, and maternity care is critical in preventing maternal suicide [45].

### 3.3. Preventive Measures to Protect Mothers

Intimate partner violence, intended as behaviors that exert power and control through the use of violence against women, occurs frequently. Perinatal care is often the only moment in the lives of many couples when there is regular contact with health care providers [46]; therefore, it could be considered an ideal “window of opportunity” to address domestic violence.

Women who are victims of physical or emotional violence by an intimate partner find it more difficult to attend prenatal appointments and are more likely to start them late. Furthermore, they may present risk behaviors during the pregnancy, such as the use of alcohol, tobacco, and illicit drugs, more frequently [47]. Efforts to ensure that women who are experiencing any kind of violence and abuse are able to access health services are needed. The possibility of an active engagement with the health care sector during the peripartum period acts as a prime occasion for screening and intervention in abused women. At the same time, the pregnancy period, characterized by more regular and frequent contacts with health professionals, results in being an ideal moment for clinicians to detect and assist victims of violence [26]. Moreover, postnatal health care visits may represent an important chance for providing support to victims, which could be built into existing governmental efforts to strenghten the quality of care, and to reduce intimate partner violence and its impact on maternal and neonatal health [27].

It has been demonstrated that antenatal psychological interventions may prevent or reduce depressed mood and anxiety or bonding failure after childbirth. In particular, early detection of interpersonal violence and advocacy interventions to encourage abused women to seek assistance, to facilitate access to useful information, and to use community resources, such as female counseling officers and protective shelters, are strongly recommended [26]. It seems fundamental to design interventions that maximize resources and minimize negative experiences such as violence during pregnancy. In such perspective, a special attention should be paid to men and their role as partners and fathers during pregnancy and the postpartum period [5]. All men, perhaps, need to better understand women’s vulnerabilities during pregnancy and the postpartum period in order to ensure collaboration, strenghten the couple, and prevent abusive behaviors or violence.

It has been recently suggested that nonspecialist providers (such as lay counselors, nurses, midwives, and teachers with no formal training in counseling interventions) may represent an effective strategy to improve access to evidence-based counseling interventions, and that integrating nonspecialist providers with evidence-based counseling interventions has the potential to address the significant burden of perinatal depression and anxiety worldwide [48].

Screening for depression and suicidality in mothers or future mothers is warranted, since both of them contribute to impairments in parenting and subsequent child development. Further studies are required to explore which scale is more suitable to assess antenatal depressive symptoms, and much attention should be paid to the mental health of pregnant women, especially those from low- and lower-middle-income countries [12]. In general, it is critical to examine associations between intimate partner violence and perinatal mental health in countries with diverse income levels.

Maternal mental health outcomes can be greatly improved by interventions incorporating home visitation, a widely used prevention approach that seeks to optimize child developmental outcomes and maternal life course by providing new mothers with education and support around child health and development of positive parenting [39]. There is evidence that home visitations decrease intimate partner violence, the rate of miscarriage, and the frequency of newborns with a low birth weight.

Supportive counseling interventions seem to be associated with a reduced number of preterm births, decreased postpartum depression, and fewer recurrent episodes of psychological and physical intimate partner violence [20]. In particular, family-based counseling plays a significant role in reducing the various types of violence against women through improving the relationship between couples during pregnancy, and family-oriented counseling contributes to deterring all forms of violence against women by increasing the awareness of couples and consequently avoiding maternal complications associated with pregnancy [49].

Although there is a debate on the effectiveness of interpersonal therapy for women with a background of intimate partner violence, recent evidence outlines that persistent social maladjustment increases the risk of depression relapse and that symptoms of post-traumatic stress disorder during pregnancy and postparum in victims of intimate partner violence can be reduced during pregnancy and immediately after delivery by sessions of interpersonal psychotherapy, focusing on enhancing interpersonal relationships, parenting skills, and social support [50].

A recent study evaluated the preliminary efficacy of a brief, motivational computer-based intervention for perinatal women victims of intimate partner violence seeking mental health treatment and found a significant reduction in emotional abuse and also reductions in physical abuse [51].

Routine screening for domestic violence has been recognized as an important identification and preventive method that allows the implementation of early intervention strategies for victims of violence. Existing screening tools should be adopted in the primary health care setting, and further research should be undertaken to determine the best means of developing and delivering domestic violence screening education and training to all multi-disciplinary health care providers [52].

Some gynecological and pregnancy-associated conditions seem to be more common in abused women (sexually transmitted infections, menstrual disorders, sexual problems, miscarriages, induced abortions) and should be considered by clinicians as red flags of possible hidden signs of abuse during routine visits [53].

## 4. Discussion

This narrative review is a descripitive-exploratory study and has several limitations: although it provides an overview of the research landscape on the important topic of interpersonal violence and peripartum mental health, offering a synthesis of the main research results in recent years, the present study does not take into account specific screening tools or tests used in the considered papers, the degree of agreement and disagreement in the results, the estimates of effect sizes, the analysis derived by different geographic regions, and the lack of a pooled estimate of interventions’ effectiveness.

It seems evident from the amount of studies included in the present review that perinatal health care needs to be integrated by the dictates of personalized medicine. Interpersonal violence during pregnancy and early motherhood requires a routine screening and peripartum may represent an optimal time when women may be seeking support to escape from violence [10]. Prenatal consultation can act as an essential tool to early dectect a case of violence and promptly promote the disruption of the cycle of abuse [9]. A very delicate moment is that of the interview with the patient. It is important for victims to know that the abuse is not their fault, and that they are not alone. Usually, pregnant women accept questions regarding interpersonal violence if there is enough privacy and confidentiality, if the setting is a private and safe environment, and if they are helped to understand that the disclosure may lead to positive consequences. The language used should not be stigmatizing or judgmental, and it is necessary to avoid making assumptions about why a patient is in a violent relationship. The response and communication should be tailored to the specific needs of the patient; for example, the person may want to focus on the well-being of their children rather than on details about intimate partner violence [54]. Explanations about the relevance of violence and abuse in pregnagncy would facilitate disclosure, and women should not be blamed or pressed into leaving the partner. It is important to encourage women to seek help and to carefully, not impulsively, consider plans to leave the marriage.

If danger exists, the patient should be immediately reffered to an appropriate agency/shelter for safety [3,55]. It seems that time represents one of the most common barriers to domestic violence screening: the time spent with the patient, the context in which screening takes place, how it is carried out, and who asks questions can influence a woman’s decision to disclose or not disclose the abuse. A possible model could be a women-centered approach to antenatal domestic and family violence screening, consisting of community-based services within a primary health care model, where midwives are educated about violence and work in partnership with local professional community agencies and hospital services [10]. Antenatal interventions for intimate partner violence and psychological care for abused pregnant women prevent bonding failure and depressive symptoms after childbirth [26].

It results in being of pivotal importance to early individuate pregnant women with poly-victimizations of intimate partner violence during pregnancy and childhood maltreatment and provide intensive and continuous support for such women to prevent maternal–infant maltreatment after childbirth [56]. Since the meaning and consequences of emotional violence vary across cultures, more in-depth knowledge of implications in the context of particular life conditions is needed, in order to facilitate targeted and culturally relevant interventions [57], and the fact that high levels of social support may partially mitigate the effect of intimate partner violence on peripartum depressive and anxiety symptoms should not be overlooked.

## 5. Conclusions

The risk of intimate partner violence during pregnancy exceeds that of the most significant obstetric risks for women [7]. There is a need to reinforce awareness about violence and its consequences in society and amongst policymakers, and to provide detailed training on intimate partner violence: specifically, the various aspects of interpersonal violence should be integrated in the curriculum in medical schools for health care professionals [42]. Discussion and assessment of all forms of interpersonal violence in a wide variety of settings will be paramount in obtaining the most accurate estimates possible. Specific socioeconomic, behavioral, and clinical characteristics of pregnant women might make them more vulnerable to violence [9]. Health care services should respond proactively to pre-pregnancy psychosocial risk factors, and interventions should be multifaceted, delineate different types of violence, control for measurement reactivity, and design a tailored intervention program adjusted to the specific needs of couples experiencing interpersonal violence [11,12,58]. All mental health care professionals referring a woman to a maternity service should ensure that they communicate complete information on any past and present mental health problem. Early identification of red flags and care pathways for women at risk for suicide is fundamental [45].

A very recent umbrella review underscored that available data and existing evidence syntheses do not capture the totality of the worldwide disease burden of intimate partner violence in pregnancy [59], and a meta-analysis conducted in Ethiopia revealed that further investigation is needed to know the reason why postpartum depression is increasing and to spread the urgency of better management [60]. Research and intervention programs should focus on several levels simoultaneously (individual, relations, community, and society): considering a single risk factor may be unsuccessful because other risk factors may persist as barriers to the desired change [46]. Supplementing screening tools with follow-up questions assessing the onset and duration of symptoms, or a complete clinical interview, would be preferred to establish reliable diagnoses and a comprehensive picture of mental health. Gaps remain in our understanding of the differential and cumulative effects of violence subtypes on mental health, and how these relations might change over time, or in our knowledge about a potential bidirectional link between intimate partner violence and perinatal mental health or comorbid mental health disorders [23].

A specific attention to the potential risk factors associated with the onset of the COVID-19 pandemic is of crucial importance to the implementation and personalization of clinical practices. Particulary during this difficult period, access to evidence-based, trauma-informed, and culturally appropriate interventions from the antenatal through postnatal periods needs priority consideration. The promotion of women awareness on the effects of pregnancy on a mental disorder and an examination of their preferences and attitudes for care are also crucial. If possible, the enhancement of family support may reduce the sense of loneliness and mitigate anxiety or depressive symptoms [61].

During the COVID-19 shelter-in-place order, mobile apps may adress the needs of pregnant women who are unwilling or unable to share their experiences of violence with their health care provider and may provide them with the opportunity to continue receiving screening and support services [62].

## Figures and Tables

**Table 1 ijerph-18-05281-t001:** Preventive measures to protect victims of interpersonal violence during peripartum.

Consider perinatal care an ideal context to address domestic violence (more regular and frequent contacts with health professionals)
Investigate risk behaviors during pregnancy (use of alcohol, tobacco, and illicit drugs)
Early detect interpersonal violence and promote advocacy interventions to encourage abused women to access information and seek assistance (female counseling officers, protective shelters)
Favor interventions of home visiting to optimize child developmental outcomes and maternal life course by providing new mothers with education and support around child health and development of positive parenting
Provide screening for depression and suicidality in mothers or future mothers
Carefully evaluate gynecological and pregnancy-associated conditions more common in abused women (sexually transmitted infections, menstrual disorders, sexual problems, miscarriages, induced abortions)
Supplement existing screening tools for depression, anxiety, and domestic violence with follow-up questions and complete clinical interviews
Accurately evaluate potential risk factors associated with the COVID-19 pandemic and guarantee access to evidence-based, trauma-informed, and culturally appropriate interventions from the antenatal through postnatal periods

## Data Availability

The data presented in this study are available on request from the corresponding author.

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
