# Peer review of "Caring for Mothers: A Narrative Review on Interpersonal Violence and Peripartum Mental Health"

_ijerph, 2021, doi:10.3390/ijerph18105281_

Round 1

Reviewer 1 Report

The article has been modified in relation to the suggestions provided and is improved and effective. I suggest publishing without further changes

Author Response

Thank you very much for your appreciation. 

Reviewer 2 Report

The paper deals with a topic of great interest, so the review offers an important synthesis of the main research results in recent years. Some modifications are suggested to improve the comprehension of the paper: a) in the structure of the paper; b) by introducing clarifications to avoid confusion generated by some ambiguous statements.

a) Regarding the structure of the work, since it is a bibliographic review, this organization is confusing:
1. Introduction
2. Methods
3. Intimate Partner Violence and Prenatal and/or Postpartum depression
4. Risk of suicide for abused mothers 
5. Preventive measures to protect mothers
6. Results
7. Discussion

Are points 3, 4 and 5 not part of the Results section of the review? A clearer structure is suggested, for example:

  1. Introduction
    2. Methods
    3. Results
        3.1. Intimate Partner Violence and Prenatal and/or Postpartum depression
        3.2. Risk of suicide for abused mothers 
        3.3. Preventive measures to protect mothers
    4. Discussion and limitations: It is important to point out in this section the limitations of the paper. For example, it does not take into account the tests used in the different papers, the degree of agreement and disagreement in the results, estimates of effect sizes, analysis by geographic regions, etc...  This affects the scope of the results of this work, and it is important to point out that this is a descriptive-exploratory study.

b) Regarding the introduction of some clarifications:

1) Lines 345-346: "In such perspective a special attention should be paid to men and their role as partners and fathers during pregnancy and the postparum period" (correct erratum). This recommendation may be valid when there is NO partner violence towards the mother. Research on intervention with batterers indicates that it is contraindicated to treat the victim and the batterer together, since it reinforces the batterer and weakens the woman. Review this statement, which can lead to important errors for the readers. The same, in table 1. This recommendation is not suitable for male batterers. 

2) Line 377: "...by sessions of interpersonal psychotherapy, focusing on enhancing interpersonal relationships, parenting skills and social support" Interpersonal psychotherapy is totally contraindicated in cases of intimate partner violence (specialized research is conclusive; it is even included in the legislation of some countries). If the reference consulted recommends it, a critical note should be included, since it is against current research in intervention with battered women.

 3) Line 447-448: "...Explanations about the relevance of violence and abuse in pregnagncy would facilitate disclosure, and women should not be blemed or pressed to leave the partner." It is important to note that women should not be blemed or pressed to leave the partner, at the moment. Readers may incorrectly interpret that women should not be supported to leave an abusive partner. Experts in intervention in this violence point out that healthy restoration of the relationship with the same person who has caused the harm is not possible. It is important to point this out so as not to lead to wrong conclusions.

Author Response

Dear Editor,

We have revised our paper carefully following reviewer’s useful suggestions. Revised parts are in red. In the present form we hope that our paper could be accepted for publication.

REVIEWER #2

The paper deals with a topic of great interest, so the review offers an important synthesis of the main research results in recent years.

Thank you for your appreciation.

Some modifications are suggested to improve the comprehension of the paper: a) in the structure of the paper; b) by introducing clarifications to avoid confusion generated by some ambiguous statements.

  1. a) Regarding the structure of the work, since it is a bibliographic review, this organization is confusing:
  2. Introduction
  3. Methods
  4. Intimate Partner Violence and Prenatal and/or Postpartum depression
  5. Risk of suicide for abused mothers
  6. Preventive measures to protect mothers
  7. Results
  8. Discussion

Are points 3, 4 and 5 not part of the Results section of the review? A clearer structure is suggested, for example:

Introduction

  1. Methods
  2. Results

    3.1. Intimate Partner Violence and Prenatal and/or Postpartum depression

    3.2. Risk of suicide for abused mothers

    3.3. Preventive measures to protect mothers

Structure of the paper has been changed as suggested.

  1. Discussion and limitations: It is important to point out in this section the limitations of the paper. For example, it does not take into account the tests used in the different papers, the degree of agreement and disagreement in the results, estimates of effect sizes, analysis by geographic regions, etc... This affects the scope of the results of this work, and it is important to point out that this is a descriptive-exploratory study.

Limitations have been added.

  1. b) Regarding the introduction of some clarifications:

  • Lines 345-346: "In such perspective a special attention should be paid to men and their role as partners and fathers during pregnancy and the postparum period" (correct erratum). This recommendation may be valid when there is NO partner violence towards the mother. Research on intervention with batterers indicates that it is contraindicated to treat the victim and the batterer together, since it reinforces the batterer and weakens the woman. Review this statement, which can lead to important errors for the readers. The same, in table 1. This recommendation is not suitable for male batterers.

It has been clarified and deleted from Table 1.

  • Line 377: "...by sessions of interpersonal psychotherapy, focusing on enhancing interpersonal relationships, parenting skills and social support" Interpersonal psychotherapy is totally contraindicated in cases of intimate partner violence (specialized research is conclusive; it is even included in the legislation of some countries). If the reference consulted recommends it, a critical note should be included, since it is against current research in intervention with battered women.

Thank you for this suggestion. It has been clarified.

  • Line 447-448: "...Explanations about the relevance of violence and abuse in pregnagncy would facilitate disclosure, and women should not be blemed or pressed to leave the partner." It is important to note that women should not be blemed or pressed to leave the partner, at the moment. Readers may incorrectly interpret that women should not be supported to leave an abusive partner. Experts in intervention in this violence point out that healthy restoration of the relationship with the same person who has caused the harm is not possible. It is important to point this out so as not to lead to wrong conclusions.

We agree that this affermation could be misinterpreted by readers. It has been clarified.

This manuscript is a resubmission of an earlier submission. The following is a list of the peer review reports and author responses from that submission.

Round 1

Reviewer 1 Report

The paper focuses on the important issue of interpersonal violence during peripartum and its consequences. The issue is also very relevant during this Covid-19 pandemic period as a risk factor to increase the isolation of women. I’ve found it complete and up-to-date

  • I suggest citing in the Introduction section a very recent paper:

Associations of intimate partner violence and financial adversity with familial homelessness in pregnant and postpartum women: A 7-year prospective study of the ALSPAC cohort. Chan CS, Sarvet AL, Basu A, Koenen K, Keyes KM. PLoS One. 2021 Jan 15;16(1):e0245507. doi: 10.1371/journal.pone.0245507. eCollection 2021.

  • Besides, as for breastfeeding avoidance, I suggest citing:

Breastfeeding avoidance following psychological intimate partner violence during pregnancy: a cohort study and multivariate analysis.

Martin-de-Las-Heras S, Velasco C, Luna-Del-Castillo JD, Khan KS.

BJOG. 2019 May;126(6):778-783. doi: 10.1111/1471-0528.15592. Epub 2019 Jan 24.

PMID: 30575266

  • About the risk of familiarity, I suggest citing this very recent paper about disorganized attachment and repetition of trauma:

Condino, V., Giovanardi, G., Vagni, M., Lingiardi, V., Pajardi, D., Colli, A.

54383140000;57193340887;56572735900;6603429203;13608492200;36134252900;

Attachment, Trauma, and Mentalization in Intimate Partner Violence: A Preliminary Investigation

(2020) Journal of Interpersonal Violence, .

https://www.scopus.com/inward/record.uri?eid=2-s2.0-85097783594&doi=10.1177%2f0886260520980383&partnerID=40&md5=e3251f9022eb7a1127c23678a49bca55

DOI: 10.1177/0886260520980383

DOCUMENT TYPE: Article

  • In relation to COVID-19, I also suggest reinforcing the need to prevent the risk of social isolation coherently with the results that the higher perceived social support decrease the depression symptoms

Mariani R, Renzi A, Di Trani M, Trabucchi G, Danskin K and Tambelli R (2020) The Impact of Coping Strategies and Perceived Family Support on Depressive and Anxious Symptomatology During the Coronavirus Pandemic (COVID-19) Lockdown.

Front. Psychiatry 11:587724.

doi: 10.3389/fpsyt.2020.587724

  • I suggest also checking some misspelling in the text

Reviewer 2 Report

Journal: IJERPH

Title: Caring for Mothers: A Narrative Review on Interpersonal Vio-2 lence and Peripartum Mental Health

Manuscript Number: IJERPH-1136104

Keywords: Depression; Perinatal women; Postpartum; Pregnancy; Violence; Personalized Medicine

Thank you for the opportunity of reading and reviewing this interesting study.

The manuscript is very well written. Nevertheless, a professional English editor should review the manuscript, which contains some minor issues regarding spelling.

Please see bellow my comments:

Abstract:

What perinatal professionals do the authors refer to in the abstract? Please clarify.

Please, provide further information in the abstract (e.g. number of papers retrieved, countries where those studies were done)

Introduction:

What is the difference between psychological and emotional abuse?

Please, provide some references to support the following paragraph: “Domestic violence is often directed at women, frequently occurring within the context of marriage or long-term relationships. Episodes of intimate partner violence in the household may create an adverse environment in the both for the caregiver experiencing the abuse both for children who are witnesses of the violence. Besides, experiences of violence usually compromise the ability and quality of parenting of the traumatized subject. Such events like conception, pregnancy, childbirth, and early transition to parenthood should be usually considered happy and joyful, but they could also represent new challenges to mother, father parent and generally to couple relationship.”

“Additive stresses”: Please clarify the definition and re-word.

In the introduction, authors wrote about femicide, intimate partner violence (in general), violence against women and violence against women during pregnancy. In my opinion, authors should only focus in what the paper is about and do not write about all those types of violence which can make the reader loose the focus of the paper.

Methods:

I cannot see the section in the manuscript, which a major limitation of this study.

There is no discussion section in the manuscript, which is an additional major limitation.

Please, authors should write the manuscript following the standard requirements to write an academic paper (in this case, a narrative review).

A table of results is more than advisable to be included.

Reviewer 3 Report

  1. Please resubmit after necessary corrections of grammatical errors etc.

Examples

Line 49: “further” instead of “futher”

Lines 69-70: “it is mostly consumed” instead of “it mostly is consumed”

Lines 93-94: Please rephrase “Eight common factors result associated with antenatal depression”

Line 226: “Above all, women” instead of “Above all women”

  1. Considering that the originality of your paper is not very clear, please clarify what novel contribution exists in your paper.

Round 2

Reviewer 2 Report

Journal: IJERPH

Title: Caring for Mothers: A Narrative Review on Interpersonal Violence and Peripartum Mental Health

Manuscript Number: IJERPH-1136104

Thank you for the opportunity of reading and reviewing the reviewed version of this manuscript.

Although the authors have tried to improve the manuscript according to this reviewer comments, the manuscript still have some major weaknesses:

  1. The Methods section is very short and do not offer enough information to replicate this study. The authors are not transparent when reporting the way they did this study.
  2. The results section appears after the discussion, which looks weird. 
  3. The results and discussion section are not well written and do not reflect their findings.
  4. I cannot see Table 1 of results.

Reviewer 3 Report

*1. Reviewer Comment*

Please resubmit after necessary corrections of grammatical errors etc.

Authors’ action

The authors corrected the errors identified.

*2. Reviewer Comment*

Considering that the originality of your paper is not very clear, please clarify what novel contribution exists in your paper.

Authors’ action

The authors mainly clarified this issue in the Abstract.

Accept in present form